# MORSE: Semantic-ally Drive-n MORpheme SEgment-er

### Abstract

We present in this paper a novel framework for morpheme segmentation which uses the morpho-syntactic regularities preserved by word representations, in addition to orthographic features, to segment words into morphemes. This framework is the first to consider vocabulary-wide syntactico-semantic information for this task. We also analyze the deficiencies of available benchmarking datasets and introduce our own dataset that was created on the basis of compositionality. We validate our algorithm across datasets and present state-of-the-art results.

## 1 Introduction

Morpheme segmentation is a core natural language processing (NLP) task used as an integral component in systems related to fields such as information retrieval (IR) (Zieman and Bleich, 1997; Kurimo et al., 2007), automatic speech recognition (ASR) (Bilmes and Kirchhoff, 2003; Kurimo et al., 2006), and machine translation (MT) (Lee, 2004; Virpioja et al., 2007). Most previous works relied solely on orthographic features (Harris, 1970; Goldsmith, 2000; Creutz and Lagus, 2002, 2005, 2007) and neglected underlying semantic information. This has led to an over-segmentation of words because a change of the surface form pattern is a necessary but insufficient indication of a morphological change. For example, although the surface form of "freshman", hints that it should be segmented to "fresh-man", "freshman" does not describe semantically the compositional meaning of "fresh" and "man".

To compensate for this lack of semantic knowledge, previous works (Schone and Jurafsky, 2000; Baroni et al., 2002; Narasimhan et al., 2015) have incorporated semantic knowledge *locally* by checking the semantic relatedness of possibly morphologically related pair of words. In (Narasimhan et al., 2015), they check for semantic relatedness using cosine similarity in word representations (Mikolov et al., 2013a; Pennington et al., 2014). A limitation of such approach is due to noise in word representations, even more so in the case of rare words. Moreover, limitation to local comparison enforces modeling morphological relations via semantic relatedness, although it has been shown that difference vectors model morphological relations more accurately (Mikolov et al., 2013b). To address this issue, we introduce a new framework (MORSE), the first to bring semantics into morpheme segmentation both on a local and a vocabulary-wide level. That is, when checking for the morphological relation between two words, we not only check for the semantic relatedness of the pair at hand (*local*), but also check if the difference vectors of pairs showing similar orthographic change are consistent (*vocabulary-wide*).

To summarize MORSE, it clusters pairs of words which only vary by an affix, for example pairs such as ("quick", "quickly") and ("hopeful", "hopefully") get clustered together. To verify the cluster of a specific affix from a semantic corpus-wide standpoint, we check for the consistency of the difference vectors (Mikolov et al., 2013b). To evaluate it from an orthographic corpus-wide perspective, we check for the size of each cluster of an affix. To evaluate each pair in a cluster locally from a semantic standpoint, we check if a pair of words in a valid affix cluster are morphologically related by checking if its difference vector is consistent with other members in the cluster and if the words in the pair are semantically related (i.e. close in the vector space). The reason for local evaluations is examples like ("on","only") which belong to the cluster of a valid affix ("ly") although

they are not morphologically related. We would expect such a pair to fail the last two local evaluation methods.

Our proposed segmentation algorithm is evaluated using benchmarking datasets from the Morpho Challenge (MC) for multiple languages and a newly introduced dataset for English which compensates for lack of discriminating capabilities in the MC dataset. Experiments reveal that our proposed framework not only outperforms the widely used approach, but also performs better than published state-of-the-art results.

The central contribution of this work is a novel framework that performs morpheme segmentation resulting in new state-of-the-art results. To the best of our knowledge this is the first unsupervised approach to consider the vocabulary-wide semantic knowledge of words and their affixes in addition to relying on their surface forms. Moreover we point out the deficiencies in the MC datasets with respect to the compositionality of morphemes and introduce our own dataset free of these deficiencies.

## 2 Related Work

Extensive work has been done in morphology learning, with tasks such as morphological analysis (Baayen et al., 1993), morphological reinflection (Cotterell et al., 2016), and morpheme segmentation. Given the less complex nature of morpheme segmentation in comparison to the other tasks, most systems developed for morpheme segmentation have been unsupervised to minimally supervised with the minimal supervision mainly used for parameter tuning.

Unsupervised morpheme segmentation could be traced back to (Harris, 1970). His work falls under the framework of Letter Successor Variety (LSV) which builds on the hypothesis that predictability of successor letters is high within morphemes and low otherwise. The most dominant pieces of work on unsupervised morpheme segmentation, Morfessor (Creutz and Lagus, 2002, 2005, 2007) and Linguistica (Goldsmith, 2000) adopt the Minimum Description Length (MDL) principle (Rissanen, 1998). In other words, they aim to minimize describing the lexicon of morphs as well as minimizing the description of an input corpus. Morfessor has a widely used API and has inspired a large body of work (Kohonen et al., 2010; Grönroos et al., 2014).

The unsupervised original implementation was later adapted (Kohonen et al., 2010; Grönroos et al., 2014) to allow for minimal supervision. Another work on minimally supervised morpheme segmentation is (Sirts and Goldwater, 2013) which relies on Adaptor Grammars (AGs) (Johnson et al., 2006). AGs learn latent tree structures over an input corpus using a nonparametric bayesian model (Sirts and Goldwater, 2013).

(Lafferty et al., 2001) use Conditional Random Fields (CRF) for morpheme segmentation. In this supervised method, the morpheme segmentation task is modeled as a sequence-to-sequence learning problem, whereby the sequence of labels defines the boundaries of morphemes (Ruokolainen et al., 2013, 2014). In contrast to the previously mentioned generative approaches of MDL and AG, this method takes a discriminative approach and allows for the inclusion of a larger set of features. In this approach, CRF learns a conditional probability of a segmentation given a word (Ruokolainen et al., 2013, 2014).

All the previously mentioned morpheme segmenters rely solely on orthographic features of morphemes. Semantics were initially introduced to morpheme segmenters in the work of (Schone and Jurafsky, 2000). They use LSA to generate word representations and then evaluate if two words are morphologically related based on semantic relatedness, as well as deterministic orthographic methods. Similarly, (Baroni et al., 2002) use edit distance and mutual information as metrics for semantic and orthographic validity of a morphological relation between two words. Recent work in (Narasimhan et al., 2015), inspired by the log-linear model in (Poon et al., 2009) incorporates semantic relatedness into the model via word representations. Other systems such as (Üstün and Can, 2016) rely solely on evaluating two words from a semantic standpoint by the use of a two-layer neural network.

Similarly, MORSE introduces semantic information into its morpheme segmenters via distributed word representations while also relying on orthographic features. Inspired by the work of (Soricut and Och, 2015), instead of merely evaluating semantic relatedness, we are the first to evaluate the morphological relationship via the difference vector of morphologically related words. Comparing the difference vectors of multiple pairs across the corpus following the same morpho-

logical relation, gives MORSE a vocabulary-wide evaluation of morphological relations learned.

## 3 System

The key limitation of previous frameworks that rely solely on orthographic features is the resulting over-segmentation. As an example, MDL-based frameworks segment "sing" to "s-ing" due to the high frequency of the morphemes: "s" and "ing". Our framework combines semantic relatedness with orthographic relatedness to eliminate such error. For the example mentioned, MORSE validates morphemes such as "s" and "ing" from an orthographic perspective, yet invalidates the relation between "s" and "sing" from a local and vocabulary-wide semantic perspective. Hence, MORSE will segment "jumping" as "jump-ing", and perform no segmentations on "sing".

To bring in semantic understanding into MORSE, we rely on word representations (Mikolov et al., 2013a; Pennington et al., 2014). These word representations capture the semantics of the vocabulary through statistics over the context in which they appear. Moreover, morphosyntactic regularities have been shown over these word representations, whereby pairs of words sharing the same relationship exhibit equivalent difference vectors (Mikolov et al., 2013b). For example, it is expected in the vector space of word representations that $\vec{w}_{\text{jumping}} - \vec{w}_{\text{jump}} \approx \vec{w}_{\text{playing}} - \vec{w}_{\text{play}}$, but $\vec{w}_{\text{sing}} - \vec{w}_{\text{s}} \not\approx \vec{w}_{\text{playing}} - \vec{w}_{\text{play}}$.

As a high level description, we first learn all possible affix transformations (morphological rules) in the language from pairs of words from an orthographic standpoint. For example, the pair ("jump", "jumping") corresponds to the valid affix transformation $\phi \xrightarrow{\text{suffix}}$ "ing", and the pair ("slow", "slogan") corresponds to the invalid rule "w" $\xrightarrow{\text{suffix}}$ "gan". Then we invalidate the rules, such as "w" $\xrightarrow{\text{suffix}}$ "gan", that do not conform to the linear relation in the vector space. We also invalidate pairs of words which, due to randomness, are orthographically related via a valid rule although they are not morphologically related, such as ("on", "only").

Now we formalize the objects we learn in MORSE and the scores (orthographic and semantic) used for validation. This constitutes the training stage. Finally, we formalize the inference stage, where we use these objects and scores to perform morpheme segmentation.

### 3.1 Training Stage

**Objects:**

- Rule set $\mathbf{R}$ made of all possible affix transformations in a language. $\mathbf{R}$ is populated via the following definition: $\mathbf{R}_{\text{suffix}} = \{\text{aff}_1 \xrightarrow{\text{suffix}} \text{aff}_2 : \exists \, (w_1, w_2) \in V^2, \text{stem}(w_1) = \text{stem}(w_2), w_1 = \text{stem}(w_1) + \text{aff}_1, w_2 = \text{stem}(w_2) + \text{aff}_2\}$, $\mathbf{R}_{\text{prefix}}$ is defined similarly for prefixes, and $\mathbf{R} = \mathbf{R}_{\text{suffix}} \cup \mathbf{R}_{\text{prefix}}$. An example $\mathbf{R}$ would be equal to $\{\phi \xrightarrow{\text{suffix}} \text{"ly"}, \phi \xrightarrow{\text{prefix}} \text{"un"}, \text{"ing"} \xrightarrow{\text{suffix}} \text{"ed"}, ...\}$.

- Support set $\mathbf{SS}_r$ for a rule $r \in \mathbf{R}$ is made of all pairs of words related via $r$ on a surface level. $\mathbf{SS}_r$ is populated via the following definition: $\mathbf{SS}_r = \{(w_1, w_2): w_1, w_2 \in V, w_1 \xrightarrow{r} w_2\}$. An example support set of the rule "ing" $\xrightarrow{\text{suffix}}$ "ed" would be $\{(\text{"playing"}, \text{"played"}), (\text{"crafting"}, \text{"crafted"}), ...\}$.

**Scores:**

- $\mathbf{score}_{\text{r\_orth}}(r)$ is a vocabulary-wide orthographic confidence score for rule $r \in \mathrm{R}$. It reflects the validity of an affix transformation in a language from an orthographic perspective. This score is evaluated as $\mathbf{score}_{\text{r\_orth}}(r) = |\mathrm{SS}_r|$.

- $\mathbf{score}_{\text{r\_sem}}(r)$ is a vocabulary-wide semantic confidence score for rule $r \in \mathrm{R}$. It reflects the validity of an affix transformation in a language from a semantic perspective. This score is evaluated as, $\mathbf{score}_{\text{r\_sem}}(r) = |\text{cluster}_r|/|\mathrm{SS}_r|^2$ where $\text{cluster}_r = \{((w_1, w_2), (w_3, w_4)): (w_1, w_2), (w_3, w_4) \in \mathrm{SS}_r, \vec{w}_1 - \vec{w}_2 \approx \vec{w}_3 - \vec{w}_4\}$. We consider $\vec{w}_1 - \vec{w}_2 \approx \vec{w}_3 - \vec{w}_4$ if $\cos(\vec{w}_4, \vec{w}_2 - \vec{w}_1 + \vec{w}_3) > 0.1$.

- $\mathbf{score}_{\text{w\_sem}}((w_1, w_2) \in \mathrm{SS}_r)$ is a vocabulary-wide semantic confidence score for a pair of words $(w_1, w_2)$. This pair of words is related via $r$ on an orthographic level, but this score reflects the validity of the morphological relation via $r$ on a semantic level. This score is evaluated as, $\mathbf{score}_{\text{w\_sem}}((w_1, w_2) \in \mathrm{SS}_r) = |\{(w_3, w_4): (w_3, w_4) \in \mathrm{SS}_r, \vec{w}_1 - \vec{w}_2 \approx \vec{w}_3 - \vec{w}_4\}|/|\mathrm{SS}_r|$. In other words, it is the fraction of pairs of words in the support set that exhibit a similar linear relation as $(w_1, w_2)$ in the vector space.

- **score**$_{\text{loc\_sem}}((w_1, w_2) \in \text{SS}_r)$ is a local semantic confidence score for a pair of words $(w_1, w_2)$. This pair of words is related via $r$ on an orthographic level, but this score reflects the semantic relatedness between the pair. This is evaluated as, **score**$_{\text{loc\_sem}}((w_1, w_2) \in \text{SS}_r)$ $= \cos(\vec{w}_1, \vec{w}_2)$.

### 3.2 Inference Stage

In this stage we perform morpheme segmentation using the knowledge gained from the first stage. We first introduce some notation. Let $\text{R}_{\text{add}} = \{r : r \in \text{R}, r = \text{aff}_1 \xrightarrow{r} \text{aff}_2, \text{aff}_1 = \phi, \text{aff}_2 \neq \phi \}$, $\text{R}_{\text{rep}} = \{r : r \in \text{R}, r = \text{aff}_1 \xrightarrow{r} \text{aff}_2, \text{aff}_1 \neq \phi, \text{aff}_2 \neq \phi \}$. In other words, we divide the rules to those where an affix is added ($\text{R}_{\text{add}}$) and to those where an affix is replaced ($\text{R}_{\text{rep}}$).

Now given a word $w$ to segment, we search for $r^*$, the solution to the following optimization problem[1]. The search space is limited to the rules that include $w$ in their support set, and thus it's a limited search space and the problem is tractable:

$$\max_{r} \quad \sum_{t_1} \text{score}_{t_1}((w', w) \in \text{SS}_r) + \sum_{t_2} \text{score}_{t_2}(\text{r})$$

$$s.\,t. \quad \text{r} \in \text{R}_{\text{add}}$$
$$\text{score}_{\text{r\_sem}}(\text{r}) > \text{t}_{\text{r\_sem}}$$
$$\text{score}_{\text{r\_orth}}(\text{r}) > \text{t}_{\text{r\_orth}}$$
$$\text{score}_{\text{w\_sem}}((w', w) \in \text{SS}_r) > \text{t}_{\text{w\_sem}}$$
$$\text{score}_{\text{loc\_sem}}((w', w) \in \text{SS}_r) > \text{t}_{\text{loc\_sem}}$$

Where $t_1 = \{\text{w\_sem, loc\_sem}\}$, $t_2 = \{\text{r\_sem, r\_orth}\}$, and $\text{t}_{\text{r\_sem}}, \text{t}_{\text{r\_orth}}, \text{t}_{\text{w\_sem}}, \text{t}_{\text{loc\_sem}}$ are hyperparameters of the system. Now given $r^* = \phi \xrightarrow{\text{suffix}} \text{suf}$, $w'$ is defined as $w' \xrightarrow{r^*} w$. Thus the system segments $w \rightarrow w'$-suf. We treat prefixes similarly. Then the system iterates over $w'$. Figure 1 shows the segmentation process of the word "unhealthy" based on the sequentially retrieved r\*.

The reason we restrict our rule set to $\text{R}_{\text{add}}$ in the optimization problem is to avoid rules such as "er" $\xrightarrow{\text{suffix}}$ "ing" like in ("player", "playing") leading to false segmentations such as "playing" $\rightarrow$ "player-ing". Yet we cannot completely restrict our search to $\text{R}_{\text{add}}$ due to rules such as "y" $\rightarrow$ "ies" in words like ("sky", "skies"). To be able to segment words such as "skies", we'd have to consider rules in $\text{R}_{\text{rep}}$ but only after searching in $\text{R}_{\text{add}}$. Thus if the first

---

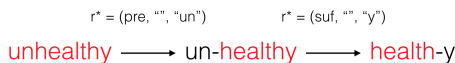

r\* = (pre, "", "un")     r\* = (suf, "", "y")

unhealthy $\longrightarrow$ un-healthy $\longrightarrow$ health-y

Figure 1: Illustration of the iterative process of segmentation in MORSE

optimization problem was unfeasible, we repeat it while replacing $\text{R}_{\text{add}}$ with $\text{R}_{\text{rep}}$. The program terminates when both optimization problems are unfeasible.

## 4 Experiments

We describe in this section the experiments done to assess the performance of MORSE. First, the performance is assessed intrinsically on the task of morpheme segmentation and against the most widely used morpheme segmenter: Morfessor. To evaluate the language agnosticity of the algorithm, we perform evaluation across three languages of varying morphology levels: English, Turkish, Finnish, with Finnish being the richest in morphology and English being the poorest. Second, we show the inadequacies of benchmarking gold datasets for this task and describe a new dataset that we create to address the inadequacy. Third, in order to highlight the effect of including semantic information, we compare MORSE against Morfessor on a set of words which should not be segmented from a semantic perspective although orthographically they seem to be segmentable (such as "freshman").

In all of our experiments (unless specified otherwise), we report precision and recall (and consequently F1 scores) with locations of morpheme boundaries being considered positives and the rest of the locations considered negatives. It should be noted that we disregard starting and ending positions of words for being trivial boundaries (Virpioja et al., 2011).

### 4.1 Setup

Both systems, Morfessor and MORSE, were trained on the same monolingual corpus: Wikipedia[2] (as of September 20, 2016) to control for affecting factors within the experiment. For each language considered, the respective Wikipedia dump was preprocessed using an available code[3]. We use Word2Vec (Mikolov et al., 2013a) to train word representations of

---

[1] $r$ and $w$ uniquely identify $w'$, and thus the search space is defined only over $r$.

[2] https://dumps.wikimedia.org
[3] https://github.com/bwbaugh/wikipedia-extractor

| Dataset | En | Fi | Tr |
|---|---|---|---|
| Tuning Data | 1000 | 1000 | 971 |
| Test Data | 686 | 760 | 809 |

Table 1: Morpho Challenge 2010 Dataset Sizes

| Word | Gold Segmentation |
|---|---|
| freshman | fresh man |
| airline | air line |
| business' | busi ness ' |
| ahead | a head |
| adultery | adult ery |

Table 2: Examples of gold morpheme segmentations from the Morpho Challenge 2010 dataset deemed invalid from a compositionality viewpoint

300 dimensions and based on a context window of size 5. Also, for computational efficiency, MORSE was limited to a vocabulary of size 1M, a restriction not enforced on Morfessor.

MORSE's hyperparameters are tuned based on a tuning set of gold morpheme segmentations. We release the source code of MORSE to the public[4].

### 4.2 Morpho Challenge Dataset

For our first intrinsic experiment we resort to the Morpho Challenge (MC) gold segmentations available online[5]. For every language, two datasets are supplied: training and development. For the purpose of our experiments, all systems use the development dataset as a test dataset, and the training dataset is used for tuning MORSE's hyperparameters. MC dataset sizes are reported in Table 1.

### 4.3 Semantically Driven Dataset

In this section we pinpoint the weaknesses of the MC dataset in assessing the relative performance of different morpheme segmenters. Consequently, we introduce a new semantically driven dataset (SD17) for morpheme segmentation along with the methodology used for creation. We make it publicly available in the canonical[6] and non-canonical[7] version (Cotterell and Vieira, 2016).

**Non-compositional segmentation:** One of the key requirements of morpheme segmentation is the compositionality of the meaning of the word from the meaning of its morphemes. This requirement is violated on multiple occasions in the MC dataset. One example from Table 2 is segmenting the word "business" into "busi-ness", which falsely assumes that "business" means the act of being busy. Such wrongly segmented words might have been truly derived from its components initially, but having undergone radical semantic change over time, they no longer semantically represent the compositionality of their components (Wijaya and Yeniterzi, 2011). Not only

does such a weakness contribute to false segmentations, but also favors segmentation methods following the MDL principle.

**Trivial instances:** The second weakness in the MC dataset is due to abundance of trivial instances. These instances lack discriminating capability since all methods can easily predict them (Baker, 2001). These instances are comprised of genetive cases (such as teacher's) as well as hyphenated words (such as turning-point). For genetive cases, segmenting at the apostrophe leads to perfect precision and recall, and thus such instances are deemed trivial. In the case of hyphenated words, segmenting at the hyphen is a correct segmentation with a very high probability. In the MC tuning dataset, in 43 times out of 46, the hyphen was a correct indication of segmentation.

**Other issues** exist in the Morpho Challenge dataset although less abundant. There are instances of **wrong segmentations** possibly due to human error. One example of such instance is "turning-point" segmented to "turning - point" instead of "turn ing - point". Another issue, which is hard to avoid, is **ambiguity** of segmentation boundaries. Take for example the word "strafed", the segmentations "straf-ed" and "strafe-d" are equally justified. In such situations, the MC dataset favors complete affixes rather than complete lemmas. This also favors MDL-based segmenters. We note that the MC dataset also provides segmentations in a canonical version such as "strafe-ed", yet for the sake of a fair comparison with Morfessor and all previously evaluated systems on the MC dataset, we consider only the former version of segmentations.

For the reasons mentioned above, we decide to create a new dataset for English gold morpheme segmentations with compositionality guiding the annotations. We select 2000 words randomly from the 10K most frequent words in the

---

[4]http://dropproxy.com/f/F94
[5]http://research.ics.aalto.fi/events/morphochallenge2010
[6]http://dropproxy.com/f/F92
[7]http://dropproxy.com/f/F93

|  | English | | | Turkish | | | Finnish | | |
|---|---|---|---|---|---|---|---|---|---|
|  | P | R | F1 | P | R | F1 | P | R | F1 |
| Morfessor | 74.46 | 56.66 | 64.35 | 40.81 | 25.00 | 31.01 | **43.09** | **28.16** | **34.06** |
| MORSE | **81.98** | **61.57** | **70.32** | **49.90** | **30.78** | **38.07** | 36.26 | 9.44 | 14.98 |

Table 3: Performance of MORSE on the MC dataset across three languages: English, Turkish, Finnish

English Wikipedia dump and have them annotated by two proficient English speakers. The segmentation criterion was to segment the word to the largest extent possible while preserving its compositionality from the segments. The inter-annotator agreement reached 91% on a word level. Based on post annotation discussions, annotators agreed on 99% of the words, and words not agreed on were eliminated along with words containing non-alpha characters to avoid **trivial instances**.

SD17 is used to evaluate the performance of Morfessor and MORSE. We claim that the performance on SD17 is a better indication of the performance of a morpheme segmenter. By the use of SD17 we expect to gain insights on the extent to which morpheme segmentation is a function of semantics in addition to orthography.

### 4.4 Handling Compositionality

One of the main claims of this paper is stating that following the MDL principle (such as Morfessor) will lead to over-segmentation. This over-segmentation happens specifically when the meaning of the word does not follow from the meaning of its morphemes. Examples include words such as "red head", "duck face", "how ever", "s ing". A subset of these words are defined by linguists as exocentric compounds (Bauer, 2008). MORSE does not suffer from this issue owing to its use of a semantic model.

We use a collection of 100 English words which appear to be segmentable but actually are not (example "however"). Such a collection will highlight a system's capability of distinguishing frequent letter sequences from the semantic contribution of this letter sequence in a word. We make this collection publicly available[8].

## 5 Results

We compare MORSE with Morfessor, and place the performance alongside the state-of-the-art published results.

---

[8]http://dropproxy.com/f/F95

|  | En | Tr | Fi |
|---|---|---|---|
| Candidate Rules | 27.5M | 14.9M | 10.8M |
| Candidate Rel. Pairs | 53.3M | 25.1M | 18.6M |

Table 4: Number of candidate rules and candidate related word pairs detected per language

### 5.1 Morpho Challenge Dataset

As demonstrated in Table 3, MORSE performs better than Mofessor on English and Turkish, and worse on Finnish. Considering English first, using MORSE instead of Morfessor, resulted in a 6% absolute increase in F1 scores. This supports our claim for the need of semantic cues in morpheme segmentation, and also validates the method used in this paper. With English being considered a less systematic language in terms of the orthographic structure of words, semantic cues are of larger need, and hence a system which relies on semantic cues is expected to perform better. Similarly, MORSE performs better on Turkish with a 7% absolute margin in terms of F1 score. On the other hand, Morfessor surpasses MORSE in performance on Finnish by a large margin as well, especially in terms of recall.

### 5.1.1 Discussion

We hypothesize that the richness of morphology in Finnish led to suboptimal performance of MORSE. This is because richness in morphology leads to word level sparsity which directly leads to: (1) Degradation of quality of word representations (2) Increased vocabulary size exacerbating the issue of limited vocabulary (recall MORSE was limited to a vocabulary of 1M). In a language with productive morphology, limiting its vocabulary results in a lower chance of finding morphologically related word pairs. This negatively impacts the training stage of MORSE which relies on the availability of such pairs. In order to detect the suffix "ly" from the word "cheerfully" MORSE needs to come across "cheerful" as well. Coming across "cheerful" is now a lower probability event

due to high sparsity. This is not as much of an issue for Morfessor under the MDL principle, since it might detect "ly" just by coming across multiple words ending with "ly" even without encountering the base forms of those words. We show how the detection of rules is affected by considering the number of candidate rules detected as well as the number of candidate morphologically related word pairs detected. As shown in Table 4, the number of detected candidate rules and candidate related words decreases with the increase in morphology in a language. This confirms our hypothesis.

This issue can be directly attributed to the limited vocabulary size in MORSE. With the increase in processing power, and thus larger vocabulary coverage, MORSE is expected to perform better.

## 5.2 Semantically Driven Dataset

The performance of MORSE and Morfessor on SD17 is shown in Table 5. The use of MC data (which does not adhere to the compositionality principle) to tune MORSE to be evaluated on SD17 (which does adhere to the compositionality principle) is not optimal. Thus, we evaluate MORSE on SD17 using 5-fold cross validation, where 80% of the dataset is used to tune and 20% is used to evaluate. Precision, Recall, and F1 scores are averaged and reported in Table 5 using code name MORSE-CV.

Based on the results in Table 5, we make the following observations. Comparing MORSE-CV to MORSE reflects the fundamental difference between SD17 and MC datasets. Knowing the basis of construction of SD17 and the fundamental weaknesses in MC datasets, we attribute the performance increase to the lack of compositionality in MC dataset. Comparing MORSE-CV to Morfessor, we observe a significant jump in performance (an increase of 24%). In comparison, the increase on the MC dataset (6%) shows that the Morpho Challenge dataset underestimates the performance gap between Morfessor and MORSE due its inherent weaknesses.

Since MORSE is equipped with the capability to retrieve full morphemes even when not present in full orthographically, a capability that Morfessor lacks, we evaluated both systems on the canonical version of SD17. The results are reported in Table 6. We notice that evaluating on the canonical form of SD17 gives a further edge for MORSE

|           | P     | R     | F1    |
|-----------|-------|-------|-------|
| Morfessor | 65.95 | 51.13 | 57.60 |
| MORSE     | 75.35 | **83.60** | 79.26 |
| MORSE-CV  | **84.6** | 78.36 | **81.29** |

Table 5: Performance of MORSE against Morfessor on the non-canonical version of SD17

|           | P     | R     | F1    |
|-----------|-------|-------|-------|
| Morfessor | 65.61 | 50.87 | 57.31 |
| MORSE     | 79.70 | 82.37 | 81.01 |
| MORSE-CV  | **85.08** | **82.90** | **83.96** |

Table 6: Performance of MORSE against Morfessor on the canonical version of SD17

over Morfessor. For evaluation on the canonical version of SD17, we switch to morpheme-level evaluation instead of boundary-level as a more suitable method for Morfessor.

We next compare MORSE against published state-of-the-art results . As one can see in Table 7 MORSE significantly performs better than published state-of-the-art results most notably (Narasimhan et al., 2015) referred to as LLSM in the Table. Comparison is also made against the top results in the latest Morpho Challenge: Morfessor S+W and Morfessor S+W+L (Kohonen et al., 2010), and Base Inference (Lignos, 2010).

## 5.3 Handling Compositionality

We compare the performance of MORSE and Morfessor on a set of words made up of morphemes which don't compose the meaning of the word. Since all the boundaries in this dataset are negative, to evaluate both systems (with MORSE tuned on SD17), we only report the number of segments generated on this dataset. The more segments a system generates, the worse is its performance.

We find that MORSE generates **7** false morphemes whereas Morfessor generates **43** false

|                 | P     | R     | F1    |
|-----------------|-------|-------|-------|
| MORSE           | **84.6** | **78.36** | **81.29** |
| LLSM            | 80.70 | 72.20 | 76.2  |
| Morfessor S+W   | 65.62 | 69.28 | 67.40 |
| Morfessor S+W+L | 67.87 | 66.43 | 67.14 |
| Base Inference  | 80.77 | 53.76 | 64.55 |

Table 7: Performance of MORSE against published state-of-the-art results

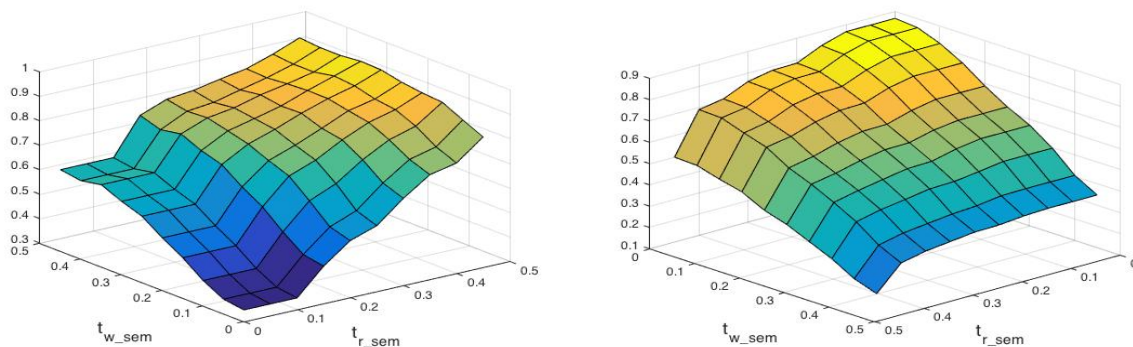

Figure 2: Precision (left) and Recall (right) of MORSE as a function of the hyperparameters: $t_{r\_sem}$, $t_{w\_sem}$

morphemes. This shows MORSE's robustness to such examples through its semantic knowledge and validates our claim that Morfessor over-segments on such examples.

## 6 Discussion

One of the benefits of MORSE against other frameworks such as MDL is its ability to identify the lemma within the segmentation. The lemma would be the last non-segmented word in the iterative process of segmentation. Hence, an advantage of our framework is its easy adaptability into a lemmatizer and even a stemmer.

Another key aspect which is not present in some of the competitive systems is the need for a small dataset for hyperparameter tuning. This is a point in favor of completely unsupervised systems such as Morfessor. On the other hand, these hyperparameters could allow for flexibility. Figure 2 shows how precision and recall changes as a function of the hyperparameter selection[9]. As one would expect, increasing the hyperparameters, in general, leads to a stricter search space and thus increases precision and decreases recall. Putting these results in perspective, the user of MORSE is given the capability of controlling for precision and recall based on the needs of the downstream task.

Moreover, to check for the level of dependency of MORSE on a set of gold morpheme segmentations for tuning, we check for the variation in performance with respect to size of tuning data. For the purpose of this experiment we take an 80-20 split of SD17 and vary the size of the tuning set. We notice that the performance (81.90% F1)

reaches a steady state at 20% ($\approx$ 300 gold segmentations) of the tuning data. This reflects the minimal dependency on gold morpheme segmentations.

As for the inference stage of MORSE, the greedy inference approach limits its performance. In other words, a wrong segmentation at the beginning will propagate and result in consequent wrong segmentations. Also, MORSE's limitation to concatenative morphology decreases its efficacy on languages that include non-concatenative morphology. This opens the stage for further research on a more optimal inference stage and a more global modeling of orthographic morphological transformations.

## 7 Conclusions and Future Work

In this paper, we have presented MORSE, a first morpheme segmenter to consider semantic structure at this scale (local and vocabulary-wide). We show its superiority over state-of-the-art algorithms using intrinsic evaluation on a variety of languages. We also pinpointed the weaknesses in current benchmarking datasets, and presented a new dataset free of these weaknesses. With a relative increase in performance reaching 24% absolute increase over Morfessor, this work proves the significance of semantic cues as well as validates a new state-of-the-art morpheme segmenter. For future work, we plan to address the limitations of MORSE: minimal supervision, greedy inference, and concatenative orthographic model. Moreover, we plan to computationally optimize the training stage for the sake of wider adoption by the community.

---

[9]Only a subset of the hyperparameters is used for display purposes

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
