# Peer review of "MORSE: Semantic-ally Drive-n MORpheme SEgment-er"

_ACL 2017 — decision unknown_

[Official Review · Reviewer 1 · rating 4 · confidence 4]
soundness 3 · originality 4 · clarity 4 · impact 4 · substance 4 · appropriateness 5 · meaningful comparison 5 · presentation format Oral Presentation

This is a nice paper on morphological segmentation utilizing word 
embeddings. The paper presents a system which uses word embeddings to 
both measure local semantic similarity of word pairs with a potential 
morphological relation, and global information about the semantic validity
of potential morphological segment types. The paper is well written and 
represents a nice extension to earlier approaches on semantically driven 
morphological segmentation.

The authors present experiments on Morpho Challenge data for three 
languages: English, Turkish and Finnish. These languages exhibit varying 
degrees of morphological complexity. All systems are trained on Wikipedia 
text. 

The authors show that the proposed MORSE system delivers clear 
improvements w.r.t. F1-score for English and Turkish compared to the well 
known Morfessor system which was used as baseline. The system fails to 
reach the performance of Morfessor for Finnish. As the authors note, this 
is probably a result of the richness of Finnish morphology which leads to 
data sparsity and, therefore, reduced quality of word embeddings. To 
improve the performance for Finnish and other languages with a similar 
degree of morphological complexity, the authors could consider word 
embeddings which take into account sub-word information. For example,

@article{DBLP:journals/corr/CaoR16,
  author    = {Kris Cao and
               Marek Rei},
  title     = {A Joint Model for Word Embedding and Word Morphology},
  journal   = {CoRR},
  volume    = {abs/1606.02601},
  year                  = {2016},
  url                 = {http://arxiv.org/abs/1606.02601},
  timestamp = {Fri, 01 Jul 2016 17:39:49 +0200},
  biburl    = {http://dblp.uni-trier.de/rec/bib/journals/corr/CaoR16},
  bibsource = {dblp computer science bibliography, http://dblp.org}
}

@article{DBLP:journals/corr/BojanowskiGJM16,
  author    = {Piotr Bojanowski and
               Edouard Grave and
               Armand Joulin and
               Tomas Mikolov},
  title     = {Enriching Word Vectors with Subword Information},
  journal   = {CoRR},
  volume    = {abs/1607.04606},
  year                  = {2016},
  url                 = {http://arxiv.org/abs/1607.04606},
  timestamp = {Tue, 02 Aug 2016 12:59:27 +0200},
  biburl    = {http://dblp.uni-trier.de/rec/bib/journals/corr/BojanowskiGJM16},
  bibsource = {dblp computer science bibliography, http://dblp.org}
}

The authors critique the existing Morpho Challenge data sets. 
For example, there are many instances of incorrectly segmented words in 
the material. Moreover, the authors note that, while some segmentations 
in the the data set may be historically valid (for example the 
segmentation of business into busi-ness), these segmentations are no 
longer semantically motivated. The authors provide a new data set 
consisting of 2000 semantically motivated segmentation of English word 
forms from the English Wikipedia. They show that MORSE deliver highly 
substantial improvements compared to Morfessor on this data set.

In conclusion, I think this is a well written paper which presents 
competitive results on the interesting task of semantically driven 
morphological segmentation. The authors accompany the submission with 
code and a new data set which definitely add to the value of the 
submission.

[Official Review · Reviewer 2 · rating 4 · confidence 5]
soundness 3 · originality 4 · clarity 5 · impact 4 · substance 4 · appropriateness 5 · meaningful comparison 4 · presentation format Oral Presentation

This paper continues the line of work for applying word embeddings for the
problem of unsupervised morphological segmentation (e.g. Soricut & Och, 2015;
Üstün & Can, 2016). The proposed method, MORSE, applies a local optimization
for segmentation of each word, based on a set of orthographic and semantic
rules and a few heuristic threshold values associated with them.

- Strengths:

The paper presents multiple ways to evaluate segmentation hypothesis on word
embeddings, and these may be useful also in other type of methods. The results
on English and Turkish data sets are convincing.

The paper is clearly written and organized, and the biliography is extensive.

The submission includes software for testing the English MORSE model and three
small data sets used in the expriments.

- Weaknesses:

The ideas in the paper are quite incremental, based mostly on the work by
Soricut & Och (2015). However, the main problems of the paper concern
meaningful comparison to prior work and analysis of the method's limitations.

First, the proposed method does not provide any sensible way for segmenting
compounds. Based on Section 5.3, the method does segment some of the compounds,
but using the terminology of the method, it considers either of the
constituents as an affix. Unsuprisingly, the limitation shows up especially in
the results of a highly-compounding language, Finnish. While the limitation is
indicated in the end of the discussion section, the introduction and
experiments seem to assume otherwise.

In particular, the limitation on modeling compounds makes the evaluation of
Section 4.4/5.3 quite unfair: Morfessor is especially good at segmenting
compounds (Ruokolainen et al., 2014), while MORSE seems to segment them only
"by accident". Thus it is no wonder that Morfessor segments much larger
proportion of the semantically non-compositional compounds. A fair experiment
would include an equal number of compounds that _should_ be segmented to their
constituents.

Another problem in the evaluations (in 4.2 and 4.3) concerns hyperparameter
tuning. The hyperparameters of MORSE are optimized on a tuning data, but
apparently the hyperparameters of Morfessor are not. The recent versions of
Morfessor (Kohonen et al. 2010, Grönroos et al. 2014) have a single
hyperparameter that can be used to balance precision and recall of the
segmentation. Given that the MORSE outperforms Morfessor both in precision and
recall in many cases, this does not affect the conclusions, but should at least
be mentioned.

Some important details of the evaluations and results are missing: The
"morpheme-level evaluation" method in 5.2 should be described or referred to.
Moreover, Table 7 seems to compare results from different evaluation sets: the
Morfessor and Base Inference methods seem to be from official Morpho Challenge
evaluations, LLSM is from Narasimhan et al. (2015), who uses aggregated data
from Morpho Challenges (probably including both development and training sets),
and MORSE is evaluated Morpho Challenges 2010 development set. This might not
affect the conclusions, as the differences in the scores are rather large, but
it should definitely be mentioned.

The software package does not seem to support training, only testing an
included model for English.

- General Discussion:

The paper puts a quite lot of focus on the issue of segmenting semantically
non-compositional compounds. This is problematic in two ways: First, as
mentioned above, the proposed method does not seem to provide sensible way of
segmenting _any_ compound. Second, finding the level of lexicalized base forms
(e.g. freshman) and the morphemes as smallest meaning-bearing units (fresh,
man) are two different tasks with different use cases (for example, the former
would be more sensible for phrase-based SMT and the latter for ASR). The
unsupervised segmentation methods, such as Morfessor, typically target at the
latter, and critizing the method for a different goal is confusing.

Finally, there is certainly a continuum on the (semantic) compositionality of
the compound, and the decision is always somewhat arbitrary. (Unfortunately
many gold standards, including the Morpho Challenge data sets, tend to be also
inconsistent with their decisions.)

Sections 4.1 and 5.1 mention the computational efficiency and limitation to one
million input word forms, but does not provide any details: What is the
bottleneck here? Collecting the transformations, support sets, and clusters? Or
the actual optimization problem? What were the computation times and how do
these scale up?

The discussion mentions a few benefits of the MORSE approach: Adaptability as a
stemmer, ability to control precision and recall, and need for only a small
number of gold standard segmentations for tuning. As far as I can see, all or
some of these are true also for many of the Morfessor variants (Creutz and
Lagus, 2005; Kohonen et al., 2010; Grönroos et al., 2014), so this is a bit
misleading. It is true that Morfessor works usually fine as a completely
unsupervised method, but the extensions provide at least as much flexibility as
MORSE has.

(Ref: Mathias Creutz and Krista Lagus. 2005. Inducing the Morphological Lexicon
of a Natural Language from Unannotated Text. In Proceedings of the
International and Interdisciplinary Conference on Adaptive Knowledge
Representation and Reasoning (AKRR'05), Espoo, Finland, June 15-17.)

- Miscellaneous:

Abstract should maybe mention that this is a minimally supervised method
(unsupervised to the typical extent, i.e. excluding hyperparameter tuning).

In section 3, it should be mentioned somewhere that phi is an empty string.

In section 5, it should be mentioned what specific variant (and implementation)
of Morfessor is applied in the experiments.

In the end of section 5.2, I doubt that increasing the size of the input
vocabulary would alone improve the performance of the method for Finnish. For a
language that is morphologically as complex, you never encounter even all the
possible inflections of the word forms in the data, not to mention derivations
and compounds.

I would encourage improving the format of the data sets (e.g.  using something
similar to the MC data sets): For example using "aa" as a separator for
multiple analyses is confusing and makes it impossible to use the format for
other languages.

In the references, many proper nouns and abbreviations in titles are written in
lowercase letters. Narasimhan et al. (2015) is missing all the publication
details.

[Official Review · Reviewer 3 · rating 4 · confidence 3]
soundness 3 · originality 4 · clarity 4 · impact 4 · substance 3 · appropriateness 5 · meaningful comparison 4 · presentation format Oral Presentation

- Strengths:
 I find the idea of using morphological compositionality to make decisions on
segmentation quite fruitful.

Motivation is quite clear

The paper is well-structured

- Weaknesses:
Several points are still unclear: 
  -- how the cases of rule ambiguity are treated (see "null->er" examples in
general discussion)
  -- inference stage seems to be suboptimal
  -- the approach is limited to known words only

- General Discussion:
The paper presents semantic-aware method for morphological segmentation. The
method considers sets of simple morphological composition rules, mostly
appearing as 'stem plus suffix or prefix'. The approach seems to be quite
plausible and the motivation behind is clear and well-argumented.

The method utilizes the idea of vector difference to evaluate semantic
confidence score for a proposed transformational rule. It's been previously
shown by various studies that morpho-syntactic relations are captured quite
well by doing word analogies/vector differences. But, on the other hand, it has
also been shown that in case of derivational morphology (which has much less
regularity than inflectional) the performance substantially drops (see
Gladkova, 2016; Vylomova, 2016). 

 The search space in the inference stage although being tractable, still seems
to be far from optimized (to get a rule matching "sky->skies" the system first
needs to searhc though the whole R_add set and, probably, quite huge set of
other possible substitutions) and limited to known words only (for which we can
there exist rules). 

 It is not clear how the rules for the transformations which are
orthographically the same, but semantically completely different are treated.
For instance, consider "-er" suffix. On one hand, if used with verbs, it
transforms them into agentive nouns, such as "play->player". On the other hand,
it could also be used with adjectives for producing comparative form, for
instance, "old->older". Or consider "big->bigger" versus "dig->digger".
More over, as mentioned before, there is quite a lot of irregularity in
derivational morphology. The same suffix might play various roles. For
instance, "-er" might also represent patiental meanings (like in "looker"). Are
they merged into a single rule/cluster? 

 No exploration of how the similarity threshold and measure may affect the
performance is presented.